# Towards Scheduling Federated Deep Learning using Meta-Gradients for Inter-Hospital Learning

## Abstract

Given the abundance and ease of access of personal data today, individual privacy has become of paramount importance, particularly in the healthcare domain. In this work, we aim to utilise patient data extracted from multiple hospital data centres to train a machine learning model without sacrificing patient privacy. We develop a scheduling algorithm in conjunction with a student-teacher algorithm that is deployed in a federated manner. This allows a central model to learn from batches of data at each federal node. The teacher acts between data centres to update the main task (student) algorithm using the data that is stored in the various data centres. We show that the scheduler, trained using meta-gradients, can effectively organise training and as a result train a machine learning model on a diverse dataset without needing explicit access to the patient data. We achieve state-of-the-art performance and show how our method overcomes some of the problems faced in the federated learning such as node poisoning. We further show how the scheduler can be used as a mechanism for transfer learning, allowing different teachers to work together in training a student for state-of-the-art performance.

## 1 Introduction

Federated learning is a field that has emerged recently due to the abundance of data available today and the risks that this poses to individuals. Privacy (particularly of personal information) is of great importance and should be protected by researchers working in machine learning. In parallel, the emergence of electronic health records (EHRs) has allowed the digitisation of much personal information pertaining to the health conditions of individuals. EHRs are often used in many machine learning research projects Shailaja et al. (2018). This often involves the transfer and storage of very sensitive information which increases the risk of data leakage. As a result, federated learning can minimise this risk by utilising the data at it's source rather than transferring it to the researchers servers to be processed.

Machine learning researchers working with EHR data will be all too familiar with the difficulty of gaining access to this information in the first place. It can be a very lengthy and exhausting process (for good reason) to gain access and utilise the EHRs of one healthcare institution let alone accessing the datasets of many. Federated learning offers an alternative in that it allows the data of the patients to be utilised while reducing the risk of their privacy being compromised.

Federated learning is not without its own limitations however. The different datasets that are stored in the different nodes may have different underlying distributions due to their data collection processes which can make machine learning across multiple domains difficult. There is also the possibility of data at each node being corrupted either maliciously or accidentally, leading to data that is undesirable to use for training. These issues can all lead to difficulties in training and convergence of the overall model being trained.

To overcome these limitations, in this work we propose the following setup. Firstly, We use federated learning to i) protect the privacy of patients by minimising the movement of their data and ii) improve the quality of our machine learning model by utilising a diverse dataset sourced from different

hospitals. As we are 'blind' to the data at the nodes, we propose the use of a student-teacher network setup. The teacher (a reinforcement learning agent) will have access to the local servers and be able to select the appropriate data for the 'student' (our model) to be trained on at that time. The 'scheduler' will be responsible for directing the teacher to a given data centre to select data for training on.

To summarise: (*The Student*:) - the machine learning model we are training. (*The Teacher*:) - a reinforcement learning agent that selects data from a data centre based on the state of the student. This essentially defines a curriculum at each training step for the student. (*The Scheduler*:) - directs the teacher to the appropriate data centre for training. This is also based on the state of the student at each training iteration.

Section 2 discusses the related work that has been carried out and Section 3 provides a detailed description of how our algorithm works. Section 4 details the datasets we benchmark our method against. We then present the results in Section 5 and discuss their significance and interesting behaviours of our model in Section 6.

## 2    RELATED WORK

Federated learning has been used by researchers to exploit larger pools of data for training Bonawitz et al. (2019), preserve data privacy Xu et al. (2019) and distributing computational resource requirements Yuan & Ma (2020). Federated learning has also been used for healthcare applications to simultaneously utilise multiple datasets to train a model on patient data.

In this work we create a model that learns in a federal fashion through the interaction of a scheduler that is trained using meta-gradients and a student-teacher algorithm that is trained using reinforcement learning.

Meta-learning has been used effectively in Such et al. (2019) where the loss of a student model on a validation set was used as a signal to update the weights of a generative model. This work demonstrated the rapid and effective training of methods that exploit meta-gradients. Meta-learning was also used in Zahavy et al. (2020), where the meta-gradients are used to tune the parameters of an actor-critic algorithm. As a result of the efficacy of this method in these domains we choose to use meta-gradients in order to schedule which data centre the gradients to update are student model will come from.

The meta-learned scheduler chooses a node representing a data centre where a student-teacher algorithm is used to sample data. Student-teacher algorithms have been used in multiple works, with the general premise that one algorithm (teacher) is trained to train another (student) Fan et al. (2018); Liu et al. (2017a). These methods have also been used with a curriculum Bengio et al. (2009), where the curriculum is either pre-defined and exploited by the teacher El-Bouri et al. (2020) or implicitly learned by the teacher during training Graves et al. (2017).

Federated learning is a method of training a model (in our case a deep neural network) by using data from multiple centres, without having central access to each of them McMahan et al. (2017). Local models at each of the data centres are iteratively updated and aggregated to form a global model. At each round of iteration, a central coordinator samples a subset, $m$, of local models, $S_m$, and sends them the current global model $G^t$. Each member of $S_m$ then updates this global model using their local data to create an updated model $L^{t+1}$. These models are then aggregated and are sent back to update the global model as:

$$G^{t+1} = G^t + \frac{\eta}{n} \sum_{i=1}^{m} \left( L_i^{t+1} - G^t \right) \tag{1}$$

where $n$ is the number of *nodes* (i.e., data centres) and $\eta$ acts as a learning rate for replacing the global model with the aggregate of the local models. While this has been shown to work in many cases, Wang et al. (2020a) make the argument that there is inherent difficulty in updating neural networks in this manner. They argue that the permutation invariance of the summation operand renders averaging in the parameter space a naive approach. For meaningful averaging to be done, the permutation must first be undone.

## 2.1 Compromising Federated Learning

One of the vulnerabilities of federated learning is that nodes being compromised can significantly affect the training of the global model Bhagoji et al. (2018). Attacks of these sort can either 'poison' the data found at one of the nodes (known as an *adversarial* attack) or bias the model that is trained at one of the nodes significantly, leading it to highly skew the aggregation step Tolpegin et al. (2020). There is also the possibility of the attack being a *single-shot attack* or a *repeated attack* Fang et al. (2020). In the single shot case, only one of the nodes is compromised whereas in the repeated case, multiple nodes can be compromised at any given time. Many works have been produced in discussing how federated learning can be compromised by introducing a backdoor into the training process Gu et al. (2017); Bagdasaryan et al. (2020). A backdoor is an attack that causes a classifier to produce unexpected behaviour if a specific trigger is added to the input space. An example is a sticker being added to an image and associating this with the incorrect label Gu et al. (2017).

Defences against these attacks have been developed with some authors using pruning of redundant neurons for the core classification task Liu et al. (2018), using outlier detection to detect potential triggers Wang et al. (2019), and re-training and preprocessing inputs Liu et al. (2017b).

In this work we aim to overcome these limitations and build defence into the training procedure through the use of a student-teacher network that actively selects which data to train on.

## 3 Methodology

Our method is comprised of three agents in the training setup, the student, the teacher and the scheduler.

## 3.1 The Overall Setup

The overall setup of our federated learning training routine is as follows. We have a scheduler that controls which node we will be learning from (this can be one-hot or we can select multiple nodes). The teacher at the node can then select a batch of data according to the state of the student. The student at the node is a copy of the global student. We use the student to forward pass the batch of data selected by the teacher and return the loss. In the one-hot scheduler scenario, we send back the loss to the global student model to update the weights via backpropagation. In the multi-node learning scenario, we aggregate the losses from all nodes selected and feed these back to the global model for updating.

## 3.2 Data Preprocessing

The first step we must take in order to exploit our teacher setup is to rank our data according to some metric. Using Wang et al. (2020b) as a guide, we choose to use the Mahalanobis distance expressed as:

$$d\left(\mathbf{x}_n\right) = \left(\left(\mathbf{x_n} - \boldsymbol{\mu}\right)^T \mathbf{S^{-1}} \left(\mathbf{x_n} - \boldsymbol{\mu}\right)\right)^{\frac{1}{2}} \tag{2}$$

for medical datasets, and the cosine similarity as our similarity metric for image datasets.

As the tabular data found in electronic health record systems consist of multiple data types, we encode these using a denoising autoencoder. This trained encoder is distributed to all the nodes so that the data in each node is processed in the same way for consistency.

## 3.3 The Teacher

For the student-teacher interaction we follow the setup in El-Bouri et al. (2020). The task of the teacher is to select a batch of data from the curriculum by selecting the index along the curriculum and the 'width' around that index to include in the selection. The following sequential steps are implemented:

- The data at each node is organised into $N$ curriculum batches according to some metric $H$.
- The teacher selects one or more batches for feeding into the student.

- A pre-trained autoencoder is used to create a latent representation of the batch.
- The student is trained on this batch and it's performance on a separate validation set is recorded.

In this work we use the Mahalanobis distance for $H$ for medical data, and cosine similarity for image data as summarised in Wang et al. (2020b).

The teacher is a reinforcement learning agent and therefore is tasked with minimising the Bellman loss function given by:

$$\mathcal{L}(\theta_i) = \left( r + \gamma \max_{a'} Q(s', a'; \theta_i^-) - Q(s, a; \theta_i) \right)^2 \tag{3}$$

where $r$ is the reward of a state-action $(s, a)$ tuple, $\gamma$ the discount factor, $Q$ is the q-value defining the value of taking an action given a state and $\theta$ and $\theta^-$ are the parameters of the prediction and target (the version of the teacher that is held constant for $K$ steps to stabilise training as described in Mnih et al. (2015)) networks respectively.

As we also choose to use an actor-critic setup for the teacher, the action space and Q-function are separately parameterised. This allows a continuous action space and the actor that selects actions is updated using the following loss:

$$\nabla_{\theta^\mu} J \approx \frac{1}{N} \sum_i \nabla_a Q\left(s, a \mid \theta^Q\right) \mid_{s=s_i, a=\mu(s_i)} \nabla_{\theta^\mu} \mu\left(s \mid \theta^\mu\right) \tag{4}$$

where $Q$ is the Q-function and $\mu$ is the policy.

The teacher can either be pre-trained, or jointly trained with the scheduler. The teacher can also either be trained on the dataset of one node and distributed to the rest or independently trained at each node. The latter is preferable due to the ability of the teacher to adapt to the dataset at hand. However, the former is useful when not all nodes in the federated system have access to computational power. The intuition is that the curriculum strategy learned by the teacher should be general for the task at hand and thereby provide strong performance.

## 3.4 The Student

The input to the teacher is the current state of the student. The student in this work is a feedforward neural network that is tasked with classification. The state of the student is defined as a representation of the weights of the student. Given a matrix of weights, $W^{ij}$, between layers $i$ and $j$ of the network, for each row, $W_n^{ij}$, we take the inner product of the row with a fixed reference vector $a$. From this inner product we extract $|\langle W_n^{ij}, a \rangle|$ and $\angle \left( W_n^{ij}, a \right)$ for $n = 1, 2, \ldots, M_i$ where $M_i$ is the number of hidden nodes in layer $i$. These values are concatenated to represent the row and this process is repeated for all rows to build the vector. For more hidden layers the process is repeated until we have one vector representing the network. This provides us with a representative vector, $\mathbf{v} \in \mathbb{R}^{2\left(\sum_l^h M_l\right)}$, where $h$ is the number of hidden layers. This vector is what is fed to the teacher to understand the state of the student.

## 3.5 The Scheduler

The scheduler is the last of our agents in the training setup. This is also a neural network that takes the student state as input and selects which of the nodes the training data should come from at the current iteration of training. This agent is trained using the meta-gradients generated from validation losses similarly to how they are employed in Such et al. (2019). We have an 'inner' loop of training whereby the student-teacher interaction takes place. In the 'outer' loop, we aggregate the losses on validation sets at each node and use this aggregated loss as the signal to update the weights of our scheduler. Figure 1 shows diagrammatically the training procedure at every iteration of training. In the outer loop the scheduler selects the node(s) to use for data selection. The global student model is sent to the node and the teacher at the node then selects the data in the inner loop and trains the student network on this. The student is then sent back to the central node and distributed to all nodes. This student is tested on separate validation sets at each node and their losses are aggregated by summing them. They are then used as the loss to update the scheduler. The inner loop loss function

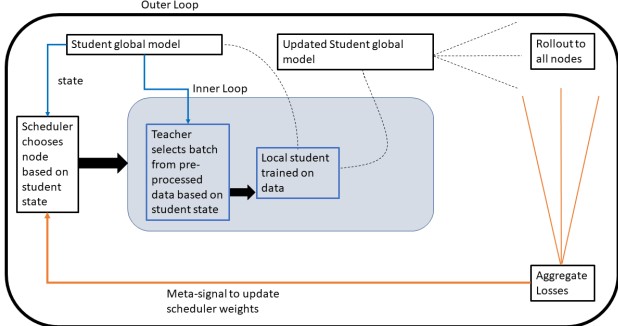

Figure 1: A diagram displaying what happens in the training routine every iteration. The black arrows indicate selection, the blue indicate the extraction of state, the dashed indicate the transfer of a model and the orange indicate the movement of losses.

is dependent upon the target task with crossentropy used for classification and mean squared error used for regression. The scheduler is updated as:

$$\theta_{sc}^{t+1} = \theta_{sc}^t + \omega \sum_{0 \leq t' \leq t} \alpha^{t-t'} \nabla \left( \mathcal{L}_{inner} \right) \tag{5}$$

where $\mathcal{L}_{inner}$ is given by:

$$\ell_{inner} \left( T \left( \mathbf{s}_t; \theta_{te} \right), \mathbf{y}_t^*; \theta_{st} \right) + \sum_{n=1}^{N-1} \ell_{inner} \left( \mathbf{x}_v^n, \mathbf{y}_v^n; \theta_{st} \right) \tag{6}$$

where $\theta_{sc}$ are scheduler weights, $\omega$ is a learning rate for stochastic gradient descent, $\alpha$ is a momentum hyperparameter, $\ell_{inner}$ is our local task loss function, $T$ is the teacher network taking as input the student state at iteration $t$, $\mathbf{s}_t$, $\mathbf{y}_t^*$ is the ground truth associated with the teacher selection and $\theta_{te}$ and $\theta_{st}$ the teacher and student parameterisations respectively. $\mathbf{x}_v^n$ and $\mathbf{y}_v^n$ are the features and labels of the validation set of node $n$ respectively.

ENTROPY LOSS

As the scheduler is trained using meta-gradients, the model may converge after a set number of iterations. While we would like convergence of the student (i.e., the task solving model), we do not necessarily need the scheduler to converge. In fact using the loss on the validation set as the signal to update the scheduler automatically prevents the scheduler from converging for very long. This is because, should the scheduler converge on selecting a particular data centre, the validation scores on the other data centres will deteriorate thereby increasing the aggregated loss on the validations and providing an update signal to the scheduler. This however was found in practice to require many iterations of training and so in order to encourage the exploration further we add an *entropic loss* term to the scheduler. This takes the form:

$$\ell_{ent} = \frac{1}{H \left[ S \left( s_t; \theta_{sc}^t \right) \right] + \epsilon} \tag{7}$$

where $H$ is entropy, $S \left( s_t; \theta_{sc}^t \right)$ is the softmax output of the scheduler and $\epsilon$ is a small positive value (we use $10^{-5}$) to prevent potential division by zero. $\ell_{ent}$ is then added on to the end of the expression shown in Equation 6 to discourage fast convergence of the scheduler.

For tabular classifications, we define the student to be a feedforward neural network consisting of 2 hidden layers and 50 nodes in each layer. These are all activated with ReLU activations. For the image recognition tasks we initialise a student that has 4 convolutional layers with 32 filters of size 3x3 in the first two layers and 64 of these in the second two. These are all activated by ReLU and maxpooled and are followed by 3 feedforward layers of size 50 nodes each.

For the results reported in Section 5, we use a pre-trained teacher (i.e., the teacher has been trained using reinforcement learning on different students for the same classification problem). The teacher

has 2 hidden layers and 150 nodes each activated by ReLU apart from the final layer (of size 2) which is activated by a tanh function. For the scheduler we use a feedforward neural network with 2 hidden layers and 100 nodes in each layer activated by ReLU. The output is activated by a softmax of size the number of nodes in the federated system.

# 4 DATASETS

In this study we considered the patient data collected in the electronic health records (EHR) of the Other Unknown Hospital (OUH), which the authors are associated with. The data is split randomly in $N$ datasets so that each can act as a separate machine in a federated system. The features include demographic, physiological and medical information (such as age, heart rate upon entry and any medical tests requested by clinical staff who greet the patient). We aim to predict which department in the hospital the patient will consume resource from (i.e., which department in the hospital will ultimately be responsible for treating the patient) rendering this a seven-class classification. In carrying out this classification, this allows hospitals to predict their resource requirements ahead of time and update scheduling and planning accordingly. Only patients who were admitted in an emergency were considered providing a dataset of 14,324 patients. A training set of 60% of the dataset was used and was balanced, leaving 8,589 patients for training on. The validation set was 20% of the dataset and testing was also 20%. These sets are then evenly divided according to the number of nodes in the federated system. The full feature set is included in the supplementary material.

## EICU

In order to validate our results on real-world data collected from different hospitals, we introduce the eICU dataset Pollard et al. (2018) also hosted on Physionet Goldberger et al. (2000). The task here is mortality prediction (binary classification) based on features extracted from admission to the ICU as is done in Sheikhalishahi et al. (2020). As this dataset contains identifiers for individual hospitals, we are able to create nodes corresponding to each hospital. The features selected are as outlined in the appendix. We choose to learn from the eight hospitals with the largest populations in the dataset leaving us with 8,594 instances. We sample 60% from each node to keep as the training set, and keep 20% for the validation set and the final 20% as the test set. As per usual, the validation set is kept on the local node for performance aggregation during the scheduler training.

## CIFAR-10

To test our methodology on the image space we also report results on the CIFAR-10 image recognition dataset Krizhevsky et al. (2009). We use 40,000 training examples for the training set and 10,000 each for validation and test set examples. We once again randomly divide the dataset into $N$ datasets to mimic a federated learning system. It should be noted that in Table 1 the CIFAR-10 dataset has been split into three samples of size 15000, 15000 and 10000 due to the possibility of the teacher selecting a full training set batchsize and memory constraints.

## MNIST

As before, we utilise the MNIST dataset LeCun et al. (1998) as another publicly available dataset to assess our results against. We use 30,000 examples for training, 10,000 for validation and 10,000 for the test set. Once again the dataset is partitioned into $N$ datasets to emulate the federated learning approach.

# 5 RESULTS

As our work lies in the intersection of two research areas within the field of machine learning (namely federated learning and student-teacher learning), we choose to use baselines from both of these fields as comparators. From the student-teacher learning side, we will assess how our method compares in terms of final model performance only. For the federated learning comparison we will compare not only the final model performance but also the robustness of the method to attack.

Table 1: Average classification accuracies and standard deviations for various baseline and state-of-the-art methods on the Ward Admission (tabular), MNIST (image) and CIFAR-10 (image) datasets. All models are averaged over the same five seeds apart from those highlighted with * which indicates that the accuracy reported from the cited text is quoted.

| METHOD | WARD ADMISSION ACC (SD) | MNIST ACC (SD) | CIFAR-10 SAMPLE 1 ACC (SD) | CIFAR-10 SAMPLE 2 ACC (SD) | CIFAR-10 SAMPLE 3 ACC (SD) | EICU AUC (SD) |
|---|---|---|---|---|---|---|
| SMBT | 0.45 (0.01) | 0.91 (0.01) | 0.65 (0.02) | 0.65 (0.02) | 0.65 (0.02) | 0.80 (0.02) |
| CURRIC | 0.48 (0.02) | 0.93 (0.02) | 0.68 (0.01) | 0.68 (0.01) | 0.68 (0.01) | 0.81 (0.01) |
| DEEPFM | 0.59 (0.01) | N/A | N/A | N/A | N/A | 0.81 (0.01) |
| DEEP+CROSSNET | 0.58 (0.02) | N/A | N/A | N/A | N/A | 0.82 (0.02) |
| DENSENET* | N/A | **0.99** (0.01) | 0.96 (0.01) | 0.96 (0.01) | 0.96 (0.01) | N/A |
| GPIPE* | N/A | **0.99** (0.01) | **0.99** (0.01) | **0.99** (0.01) | **0.99** (0.01) | N/A |
| RLST | **0.62** (0.02) | 0.95 (0.02) | 0.90 (0.01) | 0.90 (0.01) | 0.89 (0.01) | 0.85 (0.02) |
| FLST | 0.60 (0.02) | 0.94 (0.01) | 0.91 (0.01) | 0.90 (0.01) | 0.91 (0.02) | **0.86** (0.01) |

Table 2: Average classification accuracies and standard deviations for various baseline and state-of-the-art methods on the hospital ward admission (tabular), eICU (tabular), CIFAR-10 (image) and MNIST (image) datasets. All models are averaged over the same five seeds. We also show how the models perform when subjected to a data poisoning attack and a local model poisoning attack.

| METHOD | ACCURACY ACC (SD) | MODEL POISONING ACC (SD) | DATA POISONING ACC (SD) |
|---|---|---|---|
| FEDAVG (HOSPITAL) | 0.55 (0.01) | 0.33 (0.01) | 0.47 (0.02) |
| FEDMA (HOSPITAL) | 0.56 (0.02) | 0.45 (0.02) | 0.52 (0.01) |
| FLST (HOSPITAL) | **0.60** (0.02) | **0.59** (0.01) | **0.59** (0.02) |
| FEDAVG (CIFAR-10) | **0.93** (0.01) | 0.62 (0.01) | 0.65 (0.02) |
| FEDMA (CIFAR-10) | 0.93 (0.02) | 0.77 (0.02) | 0.68 (0.01) |
| FLST (CIFAR-10) | 0.90 (0.01) | **0.85** (0.01) | **0.87** (0.02) |
| FEDAVG (MNIST) | **0.95** (0.01) | 0.77 (0.01) | 0.73 (0.02) |
| FEDMA (MNIST) | **0.95** (0.01) | 0.83 (0.01) | 0.81 (0.02) |
| FLST (MNIST) | **0.95** (0.02) | **0.86** (0.02) | **0.88** (0.01) |
| FEDAVG - AUC (EICU) | 0.80 (0.01) | 0.65 (0.07) | 0.69 (0.12) |
| FEDMA - AUC (EICU) | 0.82 (0.01) | 0.71 (0.09) | 0.72 (0.04) |
| FLST - AUC (EICU) | **0.86** (0.02) | **0.80** (0.04) | **0.81** (0.03) |

FINAL MODEL PERFORMANCE

Table 1 shows how the performance of our federated learning method (FLST) compares to other state-of-the-art classification methods. The baselines we use are the reinforcement learning trained student-teacher setup without scheduling El-Bouri et al. (2020) (RLST), two state-of-the-art methods used for classifying tabular data (DeepFM Guo et al. (2017) and Deep+CrossNet Wang et al. (2017)) and two state-of-the-art classifiers for image recognition (GPipe Huang et al. (2019) and DenseNet Huang et al. (2018)). As baselines we train a standard feedforward neural network (a convolutional neural network for the image datasets) using stochastic mini-batch training (SMBT) and a curriculum (CURRIC) for comparison.

We see that our federated system is capable of producing a performance that is competing with state-of-the-art models that are trained in a centralised manner.

ROBUSTNESS OF FEDERATED TRAINING ROUTINE

Table 2 shows how our method performs when compared to other federated learning algorithms. For our baselines we use FedAvg McMahan et al. (2017) where the local models at each node are aggregated before being averaged, as well as FedMA Wang et al. (2020a), which constructs a shared

global model in a layer-wise manner by matching and averaging hidden elements (such as neurons and hidden states). We investigate how performance deteriorates when exposed to different backdoor attacks. We see that our model performs equivalently to state-of-the art federated training setups in terms of test-time performance but outperforms these models when exposed to attack. Through the use of the scheduler, our approach provides an added layer of redundancy in the system thereby allowing attacks to be avoided after their implicit detection through degraded performance on the validation sets stored at all nodes.

## 6 Implicit Defensive Setup

There are various ways in which a federated system can be attacked. This can be through the poisoning of local models at each of the nodes or poisoning of the data at each of the nodes. In the supplementary material, we outline experiments that show that the scheduler chooses appropriate nodes for training. We show how using a teaching setup, we can avoid some of the issues that could be faced by a federated learning system under attack.

## 7 Federated Learning for a Live Hospital Environment

To further assess the robustness of the proposed method, an experiment treating the hospital as a live environment is used. This is important because randomly splitting the data for each node may not be entirely characteristic of how real-world data from multiple hospital data centres would be distributed. There can be systematic differences due to the equipment that is used to take measurements, or the culture of measurement at the hospital, as well as downtime of machines meaning that recording is not possible. For this experiment, the data at two nodes are 'poisoned' in a systematic way that may mimic how data could be recorded in a real hospital environment. To achieve this, the data are separated into four nodes according to the time period during which the patient was admitted to the ED. These four nodes are patients admitted: (i) between midnight and 6 a.m., (ii) between 6 a.m. and noon, (iii) between noon and 6 p.m. and (iv) between 6 p.m. and midnight. The simulation mimics the scenario where during the morning hours, historic patient records are not available and the historic diagnosis code is therefore replaced by the generic historic diagnosis code corresponding to headache. Figure 2 shows the scheduler's selections for this setup. We observe that the scheduler

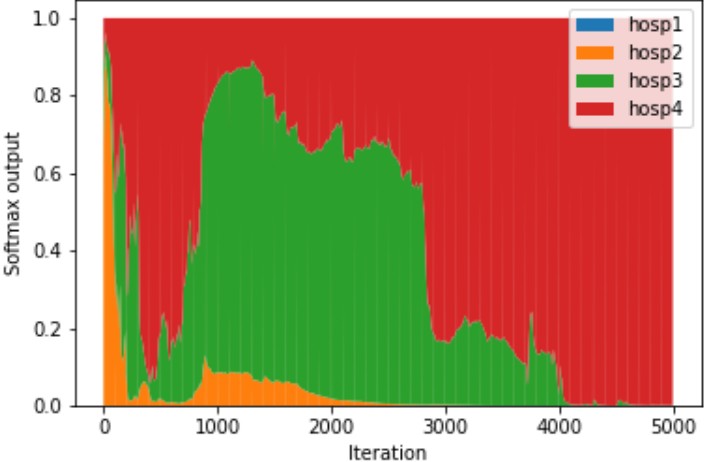

Figure 2: Scheduler selection in a system where data has been systematically poisoned according to the time it was recorded in order to simulate a database outage. Hosp 1 (node i) and 2 (node ii) represent the patients admitted in the morning hours, and 3 (node iii) and 4 (node iv) the afternoon and evening admissions respectively.

begins by using the data from the morning but rapidly changes to using afternoon data with which it

has success and therefore continues to train with. There is then a cycling between the use of afternoon and evening data while neglecting to use any data recorded during the morning. Once again, this performance is conditional on the validation set at each node being secure and unchanged. As a result, this indicates that some curation is required by a user for the validation sets to ensure that these are accurate. However, even with less curation of the training set, the scheduler may still be allowed to choose the centres from which it will gain greatest benefit for the student.

## 8 DISCUSSION AND CONCLUSION

In this work we have shown that using a federated system, with teachers at the nodes that select the data to provide gradient updates to a central model, can achieve state-of-the-art performance as well as protect the privacy of patients. We have further shown that the setup provides some protection against attacks on data stored at each node or the local models being used at each node.

However, there remain some challenges associated with this approach that need to be addressed in order for it to become practicable. The first is that the training of the centralised model (student) is inherently unstable due to the continual training that this setup expects. It is expected that the student performance will converge with training, however should data poisoning occur, the scheduler will need to continue a few iterations of training in order to recover a well-performing model. However, posing the problem in this way also provides flexibility for growing datasets at each node. All that would need to happen would be the re-sorting of the curriculum at each node and the scheduler and teachers could be used as before.

Another limitation is the need for centralised control. It is important in this setup that all the hospitals communicate their responses to the central node for actions to be taken by the scheduler. In the case of large institutions such as hospitals, this may be acceptable, but is unlikely to be for faster-paced learning environments such as learning from mobile phones, where interruptions to communication can be frequent. However, upon re-connection to the federated system, any reductions in performance to the whole system will be used as signals to improve the scheduler selection and with training the performance should recover.

Furthermore, in order to ensure diversity in selections by the scheduler we introduced the entropy loss that discouraged convergence on one selection. This may be a naive way of encouraging diversity in selection and we believe that there may be better additive losses and regularisation terms that can be used to design a loss function that will serve the purposes of the scheduler better.

For further protection against attack, sentry agents (much like the teachers) could also be trained to detect any anomalies or designed attacks within the batches selected by the teachers before the losses are passed onto the central node. This would reduce the burden of scanning the entire dataset at the node before training.

To conclude, we believe we have presented a promising direction for federated learning between large institutions such as hospitals. With further work, we believe that we can develop this into a robust system that can continually learn from growing datasets while maintaining a state-of-the-art performance for the task at hand.

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
