# OpenReview forum: "Towards Scheduling Federated Deep Learning using Meta-Gradients for Inter-Hospital Learning"
_ICLR.cc/2022/Conference — ICLR 2022 Submitted_

### Official Review · Reviewer_sDWS · 2021-10-27

**Correctness:** 2
**Technical Novelty And Significance:** 2
**Empirical Novelty And Significance:** 2
**Recommendation:** 3
**Confidence:** 4

**Main Review:**

### Strengths

- FL has been one of the most important topics in recent years. The motivation for this study to deal with data in hospitals is very understandable and compelling.
- Making FL frameworks robust against backdoor attacks is particularly an important direction. Learning a scheduler to observe clients and avoid potentially compromised ones could be a good idea (though I'm not yet perfectly convinced).
- Experimental results on new and real hospital data are invaluable.

### Weaknesses

While I found the proposed approach interesting, it is hard to properly assess the significance of the work as many technical points are unclear in the current manuscript. Specific comments are as follows:

#### (1) Teacher
I don't think that the following argument in the current manuscript is fully supposed: "The teacher can either be pre-trained or jointly trained with the scheduler. The teacher can also either be trained on the dataset of one node and distributed to the rest or independently trained at each node."

Specifically, the teacher policy specifies batch(es) of data to use as actions (Section 3.3). Then, it should be trained for each client independently since the number of data samples is not necessarily the same across clients. In other words, I cannot see when it is possible to pre-train teachers or share them among clients. Even after data are sorted based on some metrics, each data batch is essentially different among clients, or between pre-training and training data. This makes it questionable if a teacher policy trained with one dataset could be used as-is for another one.

Moreover, because the actions specify the indices of data batches to use, pre-training data and training data should have the same number of batches. Otherwise, there will be some teacher actions whose data batches do not exist (when the number of pre-training batches is larger than that of training batches) or there will be some training batches that won't be selected (when the number of pre-training batches is smaller than that of training batches). I would like to see how the proposed approach deals with these problems.

#### (2) Scheduler
While I think it interesting to learn a scheduler to select nodes, it is not clear how the scheduler is actually used to do so. As described in Section 3.5, the scheduler is a neural network with soft-max outputs with the dimension corresponding to the number of nodes. Does this mean that the scheduler selects only one node per iteration? Or how is it possible to specify multiple nodes? It also remains unclear if this approach could be scalable for hundreds of clients. Overall, the current manuscript lacks such implementation details, which should be addressed in the revised manuscript.

#### (3) Experiments
Currently, many technical details are missing in both the main paper and supplementary material. For MNIST and CIFAR-10, how was the distribution of data (or class frequency) different across nodes? Is there any data non-iidness? What specific data were used to pre-train teacher models and autoencoders? The number of clients and data non-iidness are both critical factors for federated learning. Experimental results won't be very strong if they are just about a few clients with iid data.

Moreover, I don't think that current experimental results fully demonstrate the effectiveness of the proposed approach. Since the performances of FLST and RLST (baseline with the student-teacher setup without scheduling) are almost the same, I doubt if the scheduler is working effectively. How can we confirm the effectiveness of the scheduler? Also, it’s not obvious from the results how much the teacher model contributed to the performance. I would like to see an ablation study for how the accuracy changes when only student models were trained without batch selection by the teacher.


Finally, it is currently hard to judge the validity of robustness experiments due to the lack of their setup detail. How were model poisoning and data poisoning attacks implemented? What made it difficult (or is it possible) to compare the proposed approach against the existing work on robust FL (Liu et al., 2018; Wang et al., 2019; and Liu et al., 2017b, which are just introduced briefly in the related work section)?

**Summary Of The Paper:**

This paper presents a new federated learning (FL) method augmented with "teacher" and "scheduler". The teacher is a reinforcement learning agent that observes the status of FL clients to inform which data mini-batches the clients should use to train their local model. The scheduler, on the other hand, is a meta-learner observing all the local model updates to decide which nodes (i.e., clients) to distribute the global model. Doing so by the scheduler allows the proposed method to cope with compromised clients. Experimental results using multiple hospital datasets as well as standard image datasets (MNIST and CIFAR-10) show the effectiveness of the proposed approach over some existing methods.

**Summary Of The Review:**

My initial score is reject mainly due to the lack of clarity and unconvincing experimental results. As pointed out in the weakness section, the assumptions, implementation details, effectiveness, and limitations of the teacher and the scheduler are not fully presented in the current manuscript. Although experimental results seem to show the effectiveness of the proposed approach for model/data poisoning attacks, I'm not fully convinced by the results due to the lack of their detail and the absence of existing robust FL methods. That said I would like to update my score once all these points are resolved.

---

> ### Author Response · Authors · 2021-11-23
> **Rebuttal**
>
> We thank the reviewer for their time and invaluable feedback. Please find our response to your review below:
>
> 1. We agree with the reviewer that in the best case scenario, a teacher would be trained specifically for each node. The reason we state that a teacher can be pre-trained and shared is due to the observation that curricula are able to provide a training routine that leads to smooth learning and strong performance of the network simply based on sampling according to some entropy metric as detailed in [1] and [2]. Given that the students at each node in the system are performing the same task, we make the assumption that the sampling pattern learned to maximise the performance of one student should maximise the performance of them all. The advantage of this is that it makes this method more practicable to implement in hospitals due to limited computational resource and the long training times generally needed to train reinforcement learning agents. Despite the fact that the batches between nodes will not be the same, the assumption is that due to the entropy sorting, the gradients of the batches at similar positions in the curricula between nodes should produce similar gradients (provided that the dataset is large enough that there are only a few examples in each batch). However we agree that a study verifying this should be added in future work.
>
> We agree that the number of batches in the environment should be the same between training and testing. The teacher's learning is conditioned on the weight space of the student in conjunction with the action (batch selected) which constitutes the state-action pair of the reinforcement learning setup. As a result, provided that the same number of batches is provided as an environment, the size of the individual batches can be different. If the teacher at test-time selects a batch given the student's weight space, and this produces a gradient that provides an incremental update to the student weights compared to that found in training, the teacher will select the same action given a very similar weight space. This simply means that the teacher needs to provide more updates for batches that give smaller gradients. Again the best possible performance would be with each node having it's own trained teacher, however we have found experimentally that transferring the teacher as an approximation can achieve very similar results. These results will be added to the supplementary material.
>
> 2. For simplicity in this work we chose to use the one-hot output of the scheduler network and select one node at a time. To specify multiple nodes it may be possible to use the soft outputs to make a weighted aggregation of the gradients generated by each node and update according to this. However, this method has not been implemented or experimented with yet. In terms of scalability, this method was not designed with scalability as the main priority as it is only envisioned to be used between 3 - 10 hospitals. However an ablation on the performance versus the numbers of nodes in the system has been included to analyse how the method performs with more nodes included. Implementation and technical details will be added to the supplementary material.
>
> 3. A full description of the datasets and their distributions by node will be included in the supplementary material. The data from the hospital datasets are non-iid particularly in the case of the eICU dataset where the nodes represent different hospitals and the ward admission dataset where an experiment is conducted according to the shifts (an important consideration given different shift teams have different cultures of recording data).
>
> We believe that the scheduler is working effectively as the nodes each have a subdivision of the entire training data that was used for RLST. As a result, it cannot be the case that it is simply replicating the RLST setup without scheduler input as the datasets are not the same. The ablation of using the scheduler without the teacher was conducted and found to cause oscillation in performance, with no convergence. These results will be added to the supplementary material.
>
> The reason we did not compare with these methods is that we aimed to produce an algorithm that could be used in a live environment where it is not possible to filter all the examples that are produced. Those strategies use a retrospective method to filter whereas we aimed to provide a way to train and learn how to ignore things that are not useful.
>
> We thank you once again for your excellent insights
>
>
>
>
>
>
>
> [1] - Bengio, Y., Louradour, J., Collobert, R., and Weston, J. Curriculum learning. In Proceedings of the 26th annual international conference on machine learning, pp. 41–48, 2009.
> [2] - Graves, A., Bellemare, M. G., Menick, J., Munos, R., and Kavukcuoglu, K. Automated curriculum learning for neural networks. arXiv preprint arXiv:1704.03003, 2017.

---

> > ### Comment · Reviewer_sDWS · 2021-11-23
> > **Thanks**
> >
> > Thank you very much for the detailed response!  However, I am not yet fully convinced. Further comments are below:
> >
> > > we make the assumption that the sampling pattern learned to maximise the performance of one student should maximise the performance of them all
> >
> > > due to the entropy sorting, the gradients of the batches at similar positions in the curricula between nodes should produce similar gradients
> >
> > Although they may be true for a specific problem setup studied in this work, I think that having these assumptions limits the applicability of the proposed approach. One of the key challenges in federated learning is how to make learning efficient when each client has data drawn from non-identical distributions (i.e., data non-iidness). When client data are non-iid, optimal sampling patterns should be different across clients, and gradients at similar positions in a batch won't necessarily be similar.
> >
> > Although I can't upgrade the score as long as my initial concerns were still unresolved, I think the proposed method itself is interesting, hoping that extending it to work on a more variety of problems (including those with non-iid data) would make the work stronger.

---

### Official Review · Reviewer_waUP · 2021-11-01

**Correctness:** 3
**Technical Novelty And Significance:** 2
**Empirical Novelty And Significance:** Not applicable
**Recommendation:** 5
**Confidence:** 4

**Main Review:**

This work applies federated learning with a neural scheduler to improve the effectiveness and robustness of the global model.  The idea is interesting, but they're still several concerns based on the current submission:
1. Adopt deep reinforcement learning (DRL) as a neural scheduler is not new.  In [1], the authors also adopt DRL technique to filter the data or gradient boost the training speed and performance. It seems that the main contribution of this work is to extend the DRL-based neural scheduler to the federated learning setting, which limits the novelty of the proposed work. Hence, we recommend the authors give more related works and detailed comparisons on this submission with existed (DRL-based) neural scheduler.
2. The experiments are not sufficient to demonstrate the efficacy of the proposed algorithm. In fact, there exist a large amount of work that uses hypernetwork [2] or bandit [3,4,5, etc] techniques to perform client selection.  Hence, we recommend the authors give baselines to demonstrate the efficacy of the proposed algorithm.
3. The experients demonstrate the proposed algorithm is robust to backdoor attacks. This phenomenon is interesting. However, the reason why this approach works is still unknown. We recommend the author gives more explanation on these observations either in theory or in practice.

[1]  Fan Y, Tian F, Qin T, Bian J, Liu TY. Learning what data to learn. arXiv preprint arXiv:1702.08635. 2017 Feb 28.
[2] Shamsian, A., Navon, A., Fetaya, E. and Chechik, G., 2021. Personalized Federated Learning using Hypernetworks. arXiv preprint arXiv:2103.04628.
[3] Xia W, Quek TQ, Guo K, Wen W, Yang HH, Zhu H. Multi-armed bandit-based client scheduling for federated learning. IEEE Transactions on Wireless Communications. 2020 Jul 16;19(11):7108-23.
[4] Huang T, Lin W, Shen L, Li K, Zomaya AY. Stochastic Client Selection for Federated Learning with Volatile Clients. arXiv preprint arXiv:2011.08756. 2020 Nov 17.
[5] Yoshida, N., Nishio, T., Morikura, M. and Yamamoto, K., 2020, December. MAB-based Client Selection for Federated Learning with Uncertain Resources in Mobile Networks. In 2020 IEEE Globecom Workshops (GC Wkshps (pp. 1-6). IEEE.


**Summary Of The Paper:**

In this work, the authors propose a new federated learning algorithm by adopting a neural scheduling technique. In particular, the neural scheduler is trained without needing to access the local data. Preliminary experiments demonstrate the effectiveness of the proposed algorithm.

**Summary Of The Review:**

see the comment above.

---

> ### Author Response · Authors · 2021-11-23
> **Rebuttal**
>
> We thank the reviewer for their time and invaluable feedback. Please find our response to your review below:
>
> 1.	1. We agree that deep reinforcement learning for scheduling is not a new topic. However, in this work we used this for ordering the training of mini-batches. The novelty here is the combination of this deep reinforcement learning teacher with the scheduler. Having the scheduler trained through meta-gradients provides a layer of redundancy to the sampling process. Where the teacher is constrained in needing to select a batch from the data available regardless of quality, the scheduler provides the ability for the teacher not to have to. It is this co-operation between the networks that we believe is the novelty and will be useful for researchers aiming to exploit larger quantities of sensitive data. However, we also agree that the discussion of reinforcement learning trained agents in scheduling is relevant to the work and so these discussions will be added to the related works section.
>
> 2. Whilst client selection is an important part of the work, the main aim of the algorithm is in producing as strong a performance on a held-out validation set (and ultimately test set) as possible. Given this, we felt that the appropriate baselines would be algorithms such as FedAvg and   FedMA. Including the cited works as comparison would have provided benchmarking for only one of the layers of the architecture. However, we agree that the cited works are important for verifying the efficacy of the scheduler. As a result, an ablation implementing these cited works as the schedulers for the setup will be included in the supplementary material for comparison.
>
> 3. Our hypothesis is that given the setup introduced, a backdoor attack causes a regression in the performance of networks at different nodes in the system. As a result, the degradation in reward signal is fed back to the scheduler and the network weights are updated to reflect this. This makes the scheduler less likely to select from this node than from the other nodes. However, this simulation is in the case of one node being attacked. Future work will consider how the system can be adapted to deal with multiple attacks at various nodes while maximising performance for all nodes.
>
> Thank you once again for your insights

---

> > ### Comment · Reviewer_waUP · 2021-11-25
> > **Some concerns need to be further clarified.**
> >
> > After reading the author's rebuttal, there still exist several concerns.
> > 1. From my perspective,  applying the existing deep reinforcement learning for scheduling into federated learning is actually a contribution.  However, the authors should clarify that why they adopt this type of scheduling technique for FL and what is the benefits and drawbacks, respectively.
> > 2. Client selection is a common technique for federated learning. It is better to compare additional baselines with client selection techniques.
> > 3. The reason why the proposed approach is robust to backdoor attacks is still unknown.
> >
> > Based on the current response, I would like to keep my initial score.

---

### Official Review · Reviewer_cKiw · 2021-11-02

**Correctness:** 2
**Technical Novelty And Significance:** 3
**Empirical Novelty And Significance:** 2
**Recommendation:** 5
**Confidence:** 4

**Main Review:**

Pros:
1. It is worthwhile to study the scenario of training models across different hospitals based on federated learning so that the privacy of patients is well protected.
2. Sufficient experiments are conducted to illustrate the performance of the framework as well as the effectiveness of each component.
3. Most parts of this paper are written in a logical and clear way.

Cons:
1. There is not enough theoretical contribution to either the federated learning field or the curriculum learning field. Most equations in Section 3 were proposed in previous works.
2. The authors need to point out the original novelty of this paper apart from the existing conclusions they used in the paper. Also, they need to explain how it benefits the community.
3. The experimental details are not well presented in Sections 4 and 5. For example, the hyperparameters for model training are not given in both the main text and the supplementary material.
4. The organization of this paper should be improved. For example, the details of datasets should be brief, or at least most of it should be moved to the supplementary material. Also, the discussion and conclusion should also be brief enough to summarize the most important contribution of this paper rather than explaining things in detail.


Comments:
1. The teacher component seems like a pre-trained reinforcement learning agent which is fixed during the training procedure. However, when a different student model is selected, it should correspondingly change because there will be a different input state space. This problem should be solved or further explained by the authors.
3. There are some typos: 1) In Para 4 of Section 1, “Firstly, We use“ -> ”Firstly, we use”; 2) In Para 1 of Section 4, “clinical staff who greet” -> “clinical staffs who greet”; 3) In Para 3 of Section 4, it should not be “samples” in “into three samples of”.
3. Table 1 shows the performance of decentralized FLST compared with other state-of-the-art centralized classification methods, while Table 2 shows the performance of FLST compared with other state-of-the-art decentralized federated learning methods. It can be seen that FedAvg achieves higher accuracy than most centralised methods, which is different from the results in other works. It is better for the authors to give a further explanation on this.


**Summary Of The Paper:**

This paper proposes a novel federated learning framework for model training across multiple hospital data centers. There are mainly three components in the framework including a student machine learning model, a teacher reinforcement learning agent, and a scheduler algorithm that directs the teacher to specific data centers. The setup and the details of the whole framework are well presented in the paper. Corresponding experiments are carried out to show the performance of the proposed method.

**Summary Of The Review:**

Although It is worthwhile to study the scenario of training models across different hospitals based on federated learning, there is not enough theoretical contribution to either the federated learning field or the curriculum learning field and the organization of this paper should be improved.

---

> ### Author Response · Authors · 2021-11-23
> **Rebuttal**
>
> We thank the reviewer for their time and invaluable feedback. Please find our response to your review below:
>
> Cons 1 + 2:
> Whilst we agree that the equations used have been used in previous works, we believe that the novelty of the paper is in the method in which they are employed. Utilising meta-gradients for scheduling the learning of a system from multiple sources is novel to the best of our knowledge and we believe this contribution is important for the avenues this opens for other researchers to explore. Furthermore, we believe that the experiments provided show that this approach works and can therefore encourage the building of new methodology on this basis. In terms of the benefit to the community, we believe this is very useful to practitioners of machine learning in healthcare. This is due to the many hurdles that are faced (privacy, logistical as well as security) when accessing patient data from different hospitals. As a result, we hope that this work can provide the prototype that will allow for easier learning to occur whilst keeping patient data secure so that more models can be developed with less risk to patient privacy.
>
> Cons 3:
> The experimental details will  be added to the supplementary material of the paper to give full implementation details of the setup.
>
> Cons 4:
> We agree with the organisation of the paper and we will reorganise the paper to be more succinct and move the description of the datasets to the supplementary material.
>
> Comments 1:
> We agree that one of the limitations of the method is in the need for the students to remain the same size for the teachers, due to the teacher input size. This can be remedied by training a teacher for the individual student before deploying, or jointly training the teachers, students and scheduler. Alternatively, network distillation methods could be used in conjunction with the student in order to standardise the inputs to the teacher. Ablation studies on the differing sizes of students have shown that provided the student network is not made too small (less than 30 nodes in the hidden layers) the size of the student does not affect the accuracy significantly. As a result, we believe keeping the students a standard size is a justified procedure. We will include the results of this ablation in the supplementary material.
>
> Comments 2:
> We will update the paper with corrections to the spelling errors
>
> Comments 3:
> The only instance in which FedAvg outperforms most of the baselines is in the CIFAR-10 dataset (bar the state-of-the-art image recognition networks). The reason for this is two-fold. Firstly, the CIFAR data, unlike many FedAvg benchmarks, is iid and therefore has an advantage in training when compared to other works. Secondly, for the RLST setup, due to there being a possibility of the teacher selecting the entire training set as a batch to train with (which would use up all the memory), the size of the training set had to be limited hence the three samples. As a result, FLST and RLST for CIFAR-10 have a smaller training set than the other methods. However, for the other datasets where the training sets are the same size, we see that FedAvg is usually outperformed by the benchmarks particularly in the non-iid datasets (ward admission and eICU). It should also be noted that SMBT and curric are very weak but fundamental baselines to show the performance on a very small neural network trained using standard stochastic mini-batch training and ordered batch training.
>
> Thank you once again for your great insights!

---

### Decision · Program_Chairs · 2022-01-20

**Decision:**

Reject

**Comment:**

This work describes an interesting approach of using a reinforcement learning algorithm for federated learning. The paper is well organized and the use-case of performing federated learning while preserving patient privacy is also important. However, the paper has room for improvement. Important baselines used for client selection are missing and so the deep reinforcement learning approach is not well-motivated. Many important technical details are missing such as hyperparameters and distributions for MNIST and CIFAR. The approach is also lacking novelty, DRL has been used for neural scheduling before and the authors do not suggest improvements to that. Finally, the experiments showing robustness to backdoor attacks is unconvincing and can benefit from more analysis.